# The QTL and Candidate Genes Regulating the Early Tillering Vigor Traits of Late-Season Rice in Double-Cropping Systems

**DOI:** 10.3390/ijms25031497

**Published:** 2024-01-25

**Authors:** Wei Wu, Tian-Tian Zhang, Li-Li You, Zi-Yi Wang, Si-Qi Du, Hai-Yan Song, Zao-Hai Wang, Ying-Jin Huang, Jiang-Lin Liao

**Affiliations:** 1Key Laboratory of Crop Physiology, Ecology and Genetic Breeding (Jiangxi Agricultural University), Ministry of Education of China, Nanchang 330045, China; wuwei18046805172@126.com (W.W.); songhaiyan115@163.com (H.-Y.S.);; 2Key Laboratory of Agriculture Responding to Climate Change, Jiangxi Agricultural University, Nanchang 330045, China

**Keywords:** late-season rice in double-cropping systems, QTL, BSA-seq, candidate gene, tillering vigor

## Abstract

Rice effective panicle is a major trait for grain yield and is affected by both the genetic tiller numbers and the early tillering vigor (ETV) traits to survive environmental adversities. The mechanism behind tiller bud formation has been well described, while the genes and the molecular mechanism underlying rice-regulating ETV traits are unclear. In this study, the candidate genes in regulating ETV traits have been sought by quantitative trait locus (QTL) mapping and bulk-segregation analysis by resequencing method (BSA-seq) conjoint analysis using rice backcross inbred line (BIL) populations, which were cultivated as late-season rice of double-cropping rice systems. By QTL mapping, seven QTLs were detected on chromosomes 1, 3, 4, and 9, with the logarithm of the odds (LOD) values ranging from 3.52 to 7.57 and explained 3.23% to 12.98% of the observed phenotypic variance. By BSA-seq analysis, seven QTLs on chromosomes 1, 2, 4, 5, 7, and 9 were identified using single-nucleotide polymorphism (SNP) and insertions/deletions (InDel) index algorithm and Euclidean distance (ED) algorithm. The overlapping QTL resulting from QTL mapping and BSA-seq analysis was shown in a 1.39 Mb interval on chromosome 4. In the overlap interval, six genes, including the functional unknown genes *Os04g0455650*, *Os04g0470901*, *Os04g0500600*, and ethylene-insensitive 3 (*Os04g0456900*), sialyltransferase family domain containing protein (*Os04g0506800*), and ATOZI1 (*Os04g0497300*), showed the differential expression between ETV rice lines and late tillering vigor (LTV) rice lines and have a missense base mutation in the genomic DNA sequences of the parents. We speculate that the six genes are the candidate genes regulating the ETV trait in rice, which provides a research basis for revealing the molecular mechanism behind the ETV traits in rice.

## 1. Introduction

Rice is one of the staple cereal crops that has contributed, is contributing, and will continue to contribute to global food security [1]. Rice yield is mostly affected by the factors of panicle number, grain number, and grain weight. The variation in panicle number per plant is mainly caused by the tiller number per plant, while the panicle number and genetic tiller number per plant do not have a simple linear relationship [2,3].

Rice tillering is a two-stage process: the formation of a tiller bud at each leaf axil and its subsequent outgrowth [4,5,6]. The cytoplasm-dense cells first formed a bulge at the axillary meristem in the axil of the leaf primordium, and then the bulge developed into a cone-like structure and formed into the tiller bud. During rice tiller formation, the adverse environments, including high-temperature stress, are harmful to the tiller bud growth [6,7,8], and only buds with early tillering vigor (ETV) traits could grow into rice tillers and become the final effective panicles of rice.

High temperatures in late-season rice in double-cropping systems are major adverse environmental impacts for obtaining rice tiller bud information [9]. *Japonica* rice originated from high latitude zones, and high-temperature climates are more harmful to its tiller occurrence than *indica* rice [10,11]. Recently, the cultivation pattern ‘*indica* rice to *japonica* rice’ has been implemented in a double rice cropping system in South China to ensure food security [12,13]. Recently, the cultivation pattern ‘*indica* rice *to japonica* rice’ refers to *indica*–*japonica* hybrid rice, pure *japonica* hybrid rice, and *japonica* inbred rice cultivars instead of *indica* rice cultivars planted as late-season rice in double-cropping systems [14]. Compared to *indica* rice, the grain yields of *japonica* rice are significantly higher because of the high canopy light capture capability and full use of solar radiation in late-season *japonica* rice [14,15]. However, late-season *japonica* rice at the seedling stage was just met with a high-temperature climate in the summer season when the paddy field water temperature may exceed 45 °C and sometimes reached 50 °C [9]. Under this cultivation pattern, the tiller bud growth of *japonica* rice is restrained by the paddy field hot water, which is the bottleneck for the application of the cultivation pattern in double-cropping rice systems in southern China [9,14].

Several genes and transcriptional co-regulators critical for the tiller bud normal formation in rice have been identified by genetic studies of induced mutants [16,17,18,19,20], and the molecular mechanisms behind the tiller bud normal formation were gradually clear [21,22,23,24,25]. *MONOCULM 1* (*MOC1*) is the first cloned gene and characterized as a key regulator in controlling tiller bud normal formation in rice; *MOC1* encodes a transcriptional regulator, belonging to the GRAS family nuclear protein, and *MOC1*-overexpressing plants showed higher-order tiller buds, while the loss-of-function mutant exhibits only one main culm without any tillers [4]. Two genes, *TAD1* and *TE* working upstream of *MOC1*, encode a rice homolog of *Cdh1* that functions as a co-activator or activator of APC/C (Anaphase promoting complex/cyclosome), which is a multi-subunit ubiquitin ligase; *TAD1* functions as a co-activator of APC/C to target *MOC1* for degradation in a cell-cycle-dependent manner, and *TE* recognizes and binds the APC/C to form a APC/C *TE* complex interacting with *MOC1* and *OsCDC2* to mediate the degradation of *MOC1* by the ubiquitin-26S proteasome pathway [21,22]. The degradation of *MOC1* with APC/C and TAD1/TE in a cell-cycle-dependent manner regulates rice tillering pattern [22]. On the other hand, *MOC1* is protected from degradation by binding to the DELLA protein *SLENDER RICE 1* (*SLR1*), and gibberellins trigger the degradation of *SLR1*, leading to the degradation of *MOC1* and, hence, a decrease in tiller number [23]. Recently, studies have revealed that *NUMBER PER PANICLE 6* (*GNP6*) was a novel allele of *MOC1*, and both *GNP6* and *MOC1* could not directly bind the *FLORAL ORGAN NUMBER1* (*FON1*) promoter, and functions as a co-activator of *MOC3* to activate *FON1* expression in the presence of *MOC3* [5,25]. The identified genes and the clear molecular mechanism behind tiller bud normal formation enrich the genetic resources available for improving tiller numbers in rice. However, few genes regulating the rice tiller bud to survive adversities, including hot environments and the paddy field hot water caused by high-temperature climates in the summer season, are reported.

In the present study, we identified the main QTLs and their candidate genes regulating the bud to grow into rice tillers in the summer season by carrying out QTL mapping and bulked-segregant analysis by resequencing method (BSA-seq) conjoint analysis using a rice backcross inbred line (BIL) population that generated the crossing of an *indica* rice ‘T461′ and a *japonica* rice ‘Wuyungeng24′ (WYG). T461 was derived from the hybrid progeny of Lunhui 22/Crip, and WYG was bred by the Wujin Rice Research Institute of Jiangsu Province, China.

## 2. Results

### 2.1. Phenotypic Variation in Rice Tillering Vigor

According to the investigation results, the tiller number (TN) and increased tiller number (ITN) showed significant differences between the parents T461 and WYG (Table 1), and T461 showed an ETV trait, which was earlier in emergency tillers and faster in tillering rate than those of WYG (Figure 1). The mean values of the traits TN and ITN for the BIL populations were greater than those of the parents at each time of investigation (Table 1), which suggested that the traits exhibited transgressive inheritance in the BIL populations. The absolute values of skewness and kurtosis for the TN and ITN traits in the BIL populations were less than 1, except for TN in the fourth-time investigation (T4) and ITN in the third period (P3) from the third-time investigation (T3) to the fourth-time investigation (T4).

The individual distribution among the BIL populations in TN and ITN for each time of investigation showed continuous near-normal distribution (Figure 2), which suggested that the ETV traits of rice may considered as quantitative traits and controlled by multiple genes.

### 2.2. Genetic Linkage Map Construction

After the genome polymorphism survey, 188 (13.32%) of 1276 SSR markers were almost uniformly distributed on the 12 chromosomes of rice and showed polymorphic stripes between two parents, T461 and WYG. The polymorphic SSR markers were then used and constructed a genetic linkage map (Appendix A), and the constructed genetic linkage map contains a total map distance of 2297.51cM with an average map distance of 14.01cM, covering 89.20% of the rice genome.

### 2.3. QTLs of the ETV Traits

Based on the constructed genetic linkage map and the phenotypic data generated from 537 BILs, a total of seven QTLs of the ETV traits were detected on chromosomes 1, 3, 4, and 9, with the logarithm of the odds (LOD) values ranging from 3.52 to 7.57, and explained 3.23% to 12.98% of the observed phenotypic variance with QTL IciMapping 4.2 software with the Kosambi mapping function (Table 2). Among those detected QTLs, *qETV1* was repeatedly detected twice by the phenotypic data of TN and ITN on the region of RM11797-RM11873 on chromosome 1, *qETV4-3* was detected three times on the region of RM17303-RM17445 on chromosome 4, and the other QTLs, including *qETV3-1*, *qETV3-2*, *qETV4-1*, *qETV4-2*, and *qETV9*, were detected once. The positive alleles for almost all detected QTLs were contributed by T461 to increase the ETV traits, while the QTLs *qETV3-1* and *qETV3-2* on chromosome 3 were contributed by WYG.

### 2.4. SNP/InDel Calling Via BSA-seq Method

A total of 1,159,492 confident SNPs and 149,736 confident InDels were identified and applied to determine the probable QTL region by SNP-index algorithm and Euclidean Distance (ED) algorithm, and 7 QTLs were identified by the SNP-index algorithm (Appendix A), and 5 QTLs were respectively located on chromosome 1, 2, 5, 7, and 9, and the other two QTLs were located on chromosome 4; while 13 QTLs were identified by the ED algorithm, and these QTLs were mapped to all 12 chromosomes except for chromosome 6, 8, and 10 (Appendix A). The same QTLs between the SNP-index algorithm and the ED algorithm were exactly the QTLs identified by the SNP-index algorithm, which suggested that the QTLs identified by the SNP-index algorithm were reliable, and these QTLs were named *bETV1*, *bETV2*, *bETV4-1*, *bETV4-2*, *bETV5*, *bETV7*, and *bETV9* (Table 3).

### 2.5. The Overlapping Regions

Based on the identified QTLs by QTL mapping and the BSA-seq method, we found an overlapping region with a length of 1.39 Mb at positions 22,634,622 to 24,020,000 bp on chromosome 4 based on *qETV4-2* generated from QTL mapping and *bETV4-1* from BSA-seq method (Figure 3A), and 211 unigenes were located on the region in the reference genome.

### 2.6. The Candidate Genes Revealed by RNA-seq

A total of 381.28 million raw reads generated from RNA-seq and 361.93 million clean reads were obtained after raw read filtration (Appendix A), and about 96% of high-quality reads were successfully mapped to the rice reference genome International Rice Genome Sequencing Project (IRGSP-1.0). After the unigene expression levels were normalized by the fragments per kilo bases of exon per million fragments (FPKM) value, and gene expression patterns were verified by qPCR (Appendix A), we found that only 10 unigenes of the 211 unigenes in the overlapping region showed the differential expression between the E-pool constructed by 21 rice lines with early tillering vigor traits and L-pool constructed by 18 rice lines with late tillering vigor (LTV) traits (Figure 3B).

Among the 10 differentially expressed genes (DEGs), 8 DEGs showed more than 2 fold-change upregulation, and 2 DEGs showed more than 2 fold-change downregulation in the E-pool compared to the L-pool. Gene functional annotations show that seven DEGs encode functional known proteins and three DEGs in functional unknown genes (Appendix A).

### 2.7. Missense Base Mutation for the Candidate Genes

After missense base mutation analysis for the 10 DEGs, 6 DEGs, including *Os04g0506800*, *Os04g0456900*, *Os04g0497300*, *Os04g0470901*, *Os04g0500600*, and *Os04g0455650*, showed missense base mutation, while the other 4 DEGs showed synonymous base mutation between the genomes of T461 and WYG. Genes *Os04g0506800*, *Os04g0497300*, and *Os04g0455650* had one site in single missense base mutation (Figure 4), and gene *Os04g0500600* had two sites in single missense base mutation between the genomes of the two parents T461 and WYG. Compared to the WYG genome, gene *Os04g0470901* in the T461 genome had two sites in single missense base mutation and one site in two missense base mutations, and gene *Os04g0456900* in the T461 genome had CAGAGC insert bases, which led to two amino acids, alanine (A) and arginine (R) insert. 

## 3. Discussion

### 3.1. The Candidate Genes Regulating Rice ETV Trait

The ETV is an important trait for high and stable rice yield by enhancing the effective tiller number and uniform panicle [3,26]. Therefore, identifying the genes and revealing the molecular mechanism behind the ETV trait is of great significance for high and stable yield varieties of breeding in rice. Although many QTLs and genes for tiller numbers were reported, such as the genes *OsWUS* [18], *MOC1* and *MOC3* [5], and *WUSCHEL* [27,28], few QTLs and no gene-regulating ETV traits were reported to be detected in the rice genome. In the present study, we identified an overlapping QTL which was located on chromosome 4 with a region length of 1.39 Mb by the joint analysis of the traditional QTL mapping and BSA-seq mapping methods. In the region, the functional unknown genes *Os04g0455650*, *Os04g0470901*, *Os04g0500600*, and ethylene-insensitive 3 (*Os04g0456900*), sialyltransferase family domain-containing protein (*Os04g0506800*) and ATOZI1 (*Os04g0497300*) are differentially expressed between the ETV and LTV lines of the BIL populations and have missense base mutations in the genomic DNA sequences between the parents T461 and WYG. We suggested that these six genes were the candidate genes regulating the ETV traits in rice. The results provided a research basis for revealing the molecular mechanism behind the ETV traits in rice.

### 3.2. Ethylene May Play an Important Role in ETV Traits of Rice

Plant hormones play an important role in plant growth and developmental processes [29,30,31]. Qi et al. reported that *OsEATB* restricts the ethylene-induced enhancement of gibberellin responsiveness and promotes the branching potential of both tillers and spikelets in rice [32,33]. In the present study, the ethylene signaling pathway activator (*Os04g0456900*) was also shown to be significantly upregulated based on the results of RNA-seq analysis in the rice lines with ETV traits, and the sequence alignment results showed two amino acids inserted in the T461 genome compared to WYG genome. We suggested that the mutation of gene DNA sequence in *Os04g0456900* was the internal cause of high gene expression, which enhances the ethylene signal in the rice lines with ETV traits.

### 3.3. The ST-Like Protein Unigene OsSTLP3 Plays Important Roles in Plant ETV

Sialyltransferases (STs) are widely distributed among living creatures and play a key role in a variety of physiological processes, but little is known about the existence and functions of STs in plants [34,35]. However, previous studies have demonstrated that two of three ST-like proteins genes *OsSTLP1* and *OsSTLP3* from rice have ST activity in vitro assays, and the *Arabidopsis* T-DNA insertion mutants for all of the *Arabidopsis* CSTLP homologs exhibited a lethal phenotype, suggesting that these ST-like proteins play important roles in plant development [34]. The ST-like gene mgp2 was functional in *Arabidopsis* and required for normal pollen grain germination and pollen tube growth; meanwhile, its homolog SIA2 encodes an ST-like protein that may be involved in the machinery of building RG-II and thus have a dramatic effect on the integrity of the cell wall and correct pollen tube elongation [35]. Interestingly, the ST-like protein unigene *OsSTLP3* (*Os04g0506800*) in the T461 genome in the present study had a non-synonymous mutation compared to the WYG genome and showed upregulation in the rice lines with ETV traits, which suggests that the non-synonymous mutation in *OsSTLP3* sequence probably involved in the regulation of rice ETV traits.

### 3.4. Functional Unknown Genes

After searching the databases of Rice Genome Annotation Project (RGAP; http://rice.plantbiology.msu.edu), the Rice Annotation Project Database (RAP-DB; https://rapdb.dna.affrc.go.jp/ (accessed on 17 April 2023)), and the Arabidopsis Information Resource (TAIR; https://www.arabidopsis.org/ (accessed on 17 April 2023)), genes *Os04g0455650*, *Os04g0470901*, and *Os04g0500600* are showed to be functionally unknown. Three functional unknown genes exhibited not only significant differential expression patterns between the ETV and LTV lines but also missense base mutation in genomic DNA sequences between the parents T461 and WYG. Among these unigenes, *Os04g0455650* showed a 9.06-fold upregulation in the rice lines with ETV traits compared to the rice lines with LTV traits, and a single-nucleotide substitution from G to A (G/A), resulting in Arg being replaced by His (Figure 4). *Os04g0470901* and *Os04g0500600*, respectively, were downregulated by 2.03-fold and 4.21-fold in the ETV rice lines compared to the LTV lines among the BIL populations. The DNA sequence of the unigene *Os04g0470901* showed three non-synonymous single-nucleotide substitutions; the DNA sequence of the unigene *Os04g0500600* showed double non-synonymous single-nucleotide substitution. We suggested that these functional unknown genes, *Os04g0455650*, *Os04g0470901*, and *Os04g0500600*, play important roles in rice tillers to survive the adversities of high-temperature climates and become the final effective panicles.

## 4. Materials and Methods

### 4.1. Plant Materials

The BIL populations with 537 lines (BC_2_F_13_) was developed from a cross between the *indica* rice cultivar ‘T461′ and the *japonica* rice cultivar WYG, and the flow chart of the BIL group construction is shown in Appendix A. For the parents, T461 was generated from the crossing of Lunhui 22/Cript and showed strong ETV traits, and WYG showed LTV traits and bred by Wujin Rice Research Institute of Jiangsu Province, China, when T461 and WYG were planted as late-season rice in double-cropping systems and met with a high-temperature climate in the summer season at the rice seedling stage. T461 and WYG had almost the same effective panicle number (approximately 11 effective panicles) when rice was planted as single-cropping rice and met with a mild climate in the spring season at the seedling stage of rice.

### 4.2. Field Trial and Phenotypic Evaluation

The BIL populations were planted as late-season rice of double-cropping systems in the paddy field at Jiangxi Agricultural University (28°75′ N, 115°83′ E), China. Seeds were germinated in an incubator at 37 °C for 25 h after soaking in distilled water for 48 h. Germinated seeds with normal budding were selected for each line and set of parents. The selected seeds were directly sown one seed per hill to the horizontal surface paddy field in the plot with 6 rows and 8 hills for each line, and hill space of 16.8 cm × 23.1 cm apart. Three random plots were designed for each line of the BIL population. After sowing, the paddy field soil was kept moist and non-puddled until the two-leaf stage of rice. Then, the field was irrigated and kept a 3 cm water layer, and a slow-release fertilizer (N/P/K = 17:15:15) was applied in 150 kg/hm^2^.

The rice tiller number for each plot was manually counted when 95% of BIL lines were initiated to emerge tillers. TN of all the plants (24 plants), except the plants in border rows, were counted for each plot, and the rice tiller number was counted 4 times, 1 time every 72 h. The average tiller numbers per plot were used for further analysis for the ITN.

### 4.3. Linkage Map Construction and QTL Analysis

Fresh young leaves of parents and BIL populations were separately collected from 14-day-old rice seedlings for genomic DNA extraction using a cetyltrimethylammonium bromide (CTAB) method [36]. Parental polymorphism was surveyed with 1276 simple sequence repeat (SSR) markers from the Gramene database (www.gramene.org). The phenotypic and genotypic data of 537 BILs and parents were analyzed to construct the genetic linkage map using QTL IciMapping 4.2 software with the Kosambi mapping function [37]. The LOD threshold was calculated after the 1000-permutation test (*p* < 0.05) and used to claim a putative QTL. The proportion of phenotypic variance explained (PVE) and the corresponding additive effect were estimated for each QTL.

### 4.4. BSA-seq Mapping Analysis

To improve the reliability of identified QTLs, BSA-seq analysis by high-throughput sequencing (BSA-seq) and the SNP-index algorithm and ED algorithm were further employed to screen the significant genomic region related to the ETV traits. Fresh young leaves of the 21 individuals with ETV traits and 18 individuals with LTV traits were separately collected at the tillering stage of rice for BSA-seq analysis. For BSA-seq analysis, the dominant pool (namely the ETV traits pool, E-pool) was constructed by mixing an equal amount of leaf tissues from 21 individuals with ETV traits, while the recessive pool (namely the LTV traits pool, L-pool) was constructed by 18 individuals with LTV traits from the BIL populations, respectively. After library sequencing, high-quality read aligning and SNP calling, the SNP-index algorithm, and the ED algorithm were used to identify the candidate regions for the ETV traits, according to the previous method [38]. The same overlapping regions identified by the SNP-index algorithm and the ED algorithm were considered as BSA-seq mapping the candidate regions related to the ETV traits.

### 4.5. Analysis of the Overlapping Regions

The overlapping regions related to the ETV traits identified by QTL mapping and BSA-seq were manually obtained according to the intervals of the physical location on rice chromosomes, and the gene DNA sequence information for all the genes in the overlapping regions was retrieved from the database of IRGSP-1.0 (https://rgp.dna.affrc.go.jp/E/IRGSP/ (accessed on 6 May 2023)) for further candidate gene expression pattern and missense base mutation analysis.

### 4.6. Expression Pattern Analysis for the Genes in the Overlapping Regions

To further screen the candidate genes, transcriptome analysis by the RNA-sequencing method was employed to detect the gene expression patterns of all the genes in the overlapping regions. The BIL populations were cultivated as late-season rice in double-cropping systems. When 95% of BIL lines initiated to emerge tillers, tiller nodes of the 21 ETV and 18 LTV lines used to construct E-pool and L-pool in BSA-seq analysis were consecutively collected 4 times, 1 time every 72 h. Eight tiller nodes from each line were separately collected. Equal tillering nodes from each ETV rice line were mixed to form a T-Mix sample, and equal tillering nodes from each LTV rice line formed an S-Mix sample. Four biological replicates were carried out. After total RNA extraction, quality inspection, and concentration determination for each mixed sample, sequencing libraries were constructed using NEBNext^®^ Ultra™ RNA Library Prep Kit for Illumina^®^ and sequenced on the Illumina sequencing platform by Genedenovo Biotechnology, Co., Ltd. (Guangzhou, China).

The high-quality reads were then mapped to the rice reference genome IRGSP-1.0 after raw read filtration. The number of reads mapped to each gene was counted, and the unigene expression levels were normalized by FPKM value [39]. DEGs between T-Mix samples and S-Mix samples were screened using a preset threshold |log_2_fold change| ≥ 2 and false discovery rate (FDR) ≤ 0.05. The DEGs in the overlapping regions were DNA sequence retrieved and verified by qPCR in expression pattern among the parents T461 and WYG and functional annotated in the Rice Genome Annotation Project (RGAP; http://rice.plantbiology.msu.edu (accessed on 17 May 2023)), the Rice Annotation Project Database (RAP-DB; https://rapdb.dna.affrc.go.jp/ (accessed on 17 May 2023)), and the Arabidopsis Information Resource (TAIR; https://www.arabidopsis.org/ (accessed on 19 May 2023)).

### 4.7. Analysis of Missense Base Mutation for the DEGs

According to the genomic DNA sequences retrieved from the database IRGSP-1.0, the specific primers for the DEGs were, respectively, designed by Primer 5. Fresh young leaves of two parents, T461 and WYG, were separately collected from 14-day-old rice seedlings, and genomic DNA extraction was conducted using the CTAB method. The DNA for DEGs were, respectively, amplified by PCR from the genomes of two parents T461 and WYG, and the sequence message for the DEGs was obtained by Sanger method sequencing using PCR production. Analysis of missense base mutation in the genomic DNA sequences between the parents T461 and WYG was carried out for the DEGs by the software DNAMAN 9.0.

## Figures and Tables

**Figure 1 ijms-25-01497-f001:**
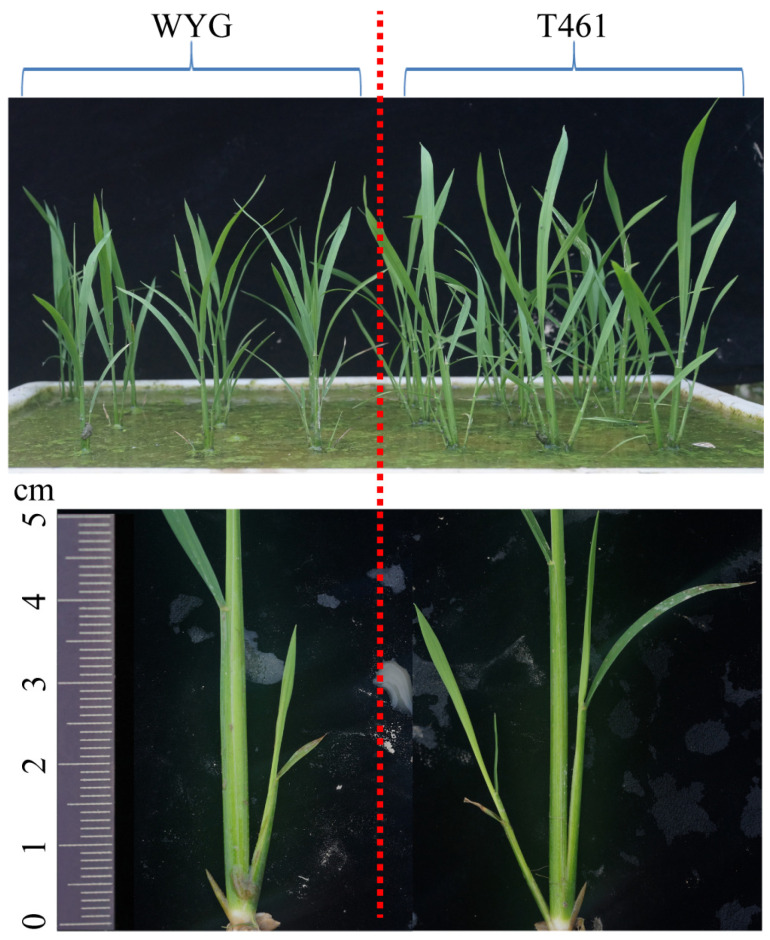
The difference in tiller numbers of the parents T461 and WYG for the rice backcross inbred line (BIL) populations. T461 and WYG are the parents of the BIL populations, T461 is the *indica* rice cultivar, and WYG is the *japonica* rice cultivar ‘Wuyungeng 24’. T461 and WYG were cultivated as late-season rice in double-cropping systems in Nanchang city, Jiangxi province, China.

**Figure 2 ijms-25-01497-f002:**
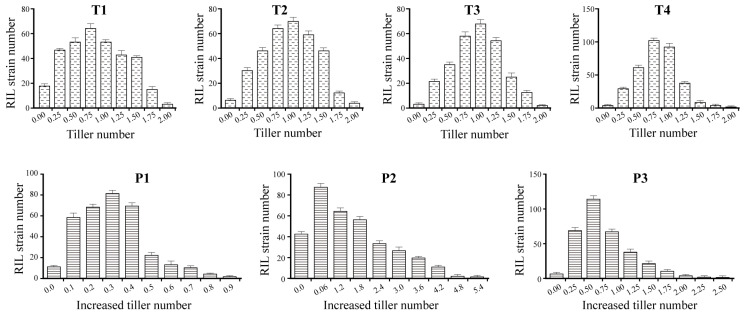
The individual distribution among the BIL populations in TN and ITN. BIL indicates the recombinant inbred line of rice. TN and ITN indicate tiller number and increased tiller number, respectively. T1, T2, T3 and T4 indicate the 1st-, 2nd-, 3rd-, and 4th-time investigation for TN, respectively. P1 indicates the ITN from the 1st-time to the 2nd-time investigation for TN, P2 indicates the ITN from the 2nd-time to the 3rd-time investigation for TN, and P3 indicates the ITN from the 3rd-time to the 4th-time investigation for TN.

**Figure 3 ijms-25-01497-f003:**
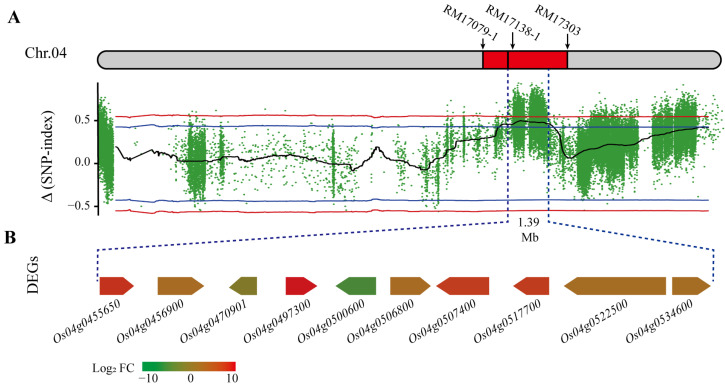
The overlapping region on chromosome 4 and the differential expressed genes (DEGs) in the region. (**A**) The overlapping region 1.39 Mb among the makers RM17079-1, RM17138-1, and RM17303 was identified by QTL mapping and the BSA-seq method. The colored points indicate the calculated ∆(SNP-index) values. The red and blue lines indicate thresholds with confidence levels of 0.99 and 0.95, respectively. Chr.04 indicates the rice chromosome 4. (**B**) DEGs indicate the differential expressed genes between the E-pool constructed by 21 rice lines with early tillering vigor traits and L-pool constructed by 18 rice lines with late tillering vigor traits. FC indicates fold change in the gene expression, and different colors indicate different fold changes in Log_2_FC values in gene expression.

**Figure 4 ijms-25-01497-f004:**
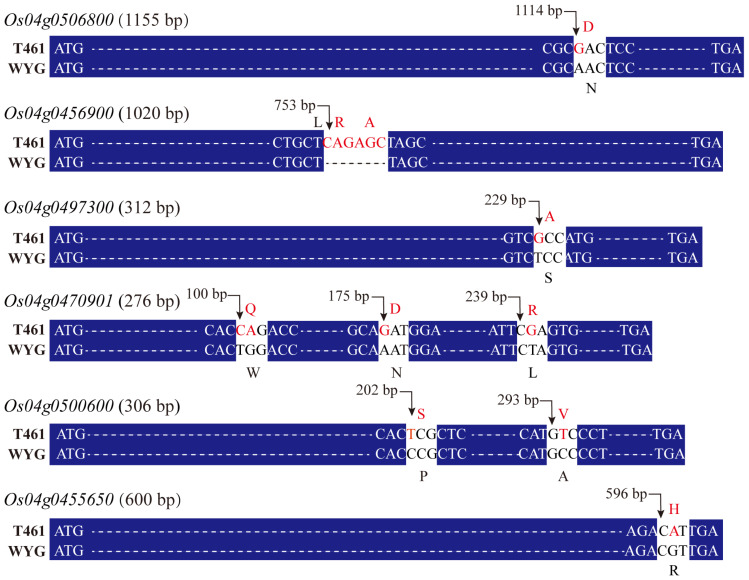
The missense base mutation of DNA sequence for the DEGs. T461 indicates the DNA sequence of the indica rice cultivar ‘T461’, and WYG indicates the DNA sequence of the japonica rice cultivar ‘Wuyungeng 24’. Nucleotide base symbols in red font indicate the missense base mutation between the T461 and WYG genomes.

**Table 1 ijms-25-01497-t001:** Statistical description for tillering traits of the BIL populations and their parents.

Traits	Time	Parents				BIL Populations ^6^					
		T461	WYG	*t*-Test		Mean	SD	Min	Max	Kurtosis	Skewness
TN ^1^	T1 ^3^	0.47	0.12	*** ^5^		0.85	0.48	0	1.93	−0.88	0.17
	T2	0.61	0.15	***		1.15	0.52	0.04	2.53	−0.68	0.02
	T3	0.75	0.21	***		1.30	0.56	0.08	2.80	−0.55	0.01
	T4	1.89	0.87	***		1.95	0.78	0.10	5.52	1.43	0.52
ITN ^2^	P1 ^4^	0.14	0.03	***		0.30	0.17	0	0.86	0.41	0.68
	P2	0.14	0.06	***		0.15	0.12	0	0.54	−0.08	0.77
	P3	1.14	0.66	***		0.66	0.42	0	2.90	4.66	1.64

^1^ TN indicates tiller number; ^2^ ITN indicates increased tiller number; ^3^ T1, T2, T3 and T4 indicate the 1st, 2nd, 3rd and 4th times of tiller number investigation, respectively; ^4^ P1 indicates the increased tiller number during the period from the 1st times (T1) to the 2nd time (T2) of investigation, and P2 indicates the increased tiller number during the period from T2 to T3, and P3 indicates the increased tiller number during the period from T3 to T4; ^5^ *** indicates significant difference at the probability level of 1‰ between two parents, according to Student’s *t*-test; ^6^ backcross inbred lines (BIL) populations include 537 lines, and planted as late-season rice in double-cropping systems and designed in triplicate for each line.

**Table 2 ijms-25-01497-t002:** QTLs of ETV traits detected by composite interval mapping.

QTL	Chr. ^2^	Time	Marker Interval	CI ^3^ (cM)	Physical Location (Mb)	LOD ^4^	PVE ^5^ (%)	Add ^6^
*qETV1* ^1^	1	T4	RM11797-RM11873	216.2–219.5	34.52–36.43	3.60	4.23	0.17 ^7^
	1	P3	RM11797-RM11873	214.5–219.5	34.52–36.43	6.18	7.96	0.11
*qETV3-1*	3	P1	RM15861-RM15795	120.5–134.5	29.99–28.81	4.11	4.50	−0.04
*qETV3-2*	3	P2	RM15795-RM15711	127.5–136.5	28.81–27.60	3.52	3.23	−0.03
*qETV4-1*	4	P2	RM17079_1-RM17138	2.5–19.5	22.63–24.02	4.26	6.38	0.04
*qETV4-2*	4	T3	RM17138_1-RM17303	24.5–37.5	24.02–27.27	7.57	12.98	0.18
*qETV4-3*	4	T2	RM17303-RM17445	38.5–52.5	27.27–30.36	4.67	6.02	0.16
	4	T4	RM17303-RM17445_1	38.5–53.5	27.27–30.36	3.76	6.15	0.20
	4	P1	RM17303_1-RM17445	42.5–58.5	27.27–30.36	4.60	6.52	0.05
*qETV9*	9	P3	RM24631-RM24331_1	125.5–137.5	19.79–21.41	4.94	9.42	0.13

^1^ ETV indicates early tillering vigor; ^2^ Chr. indicates the chromosomes of rice; ^3^ CI indicates confidence interval; ^4^ LOD indicates logarithm of the odd score for each additive QTL, and LOD > 3.517 was the critical standard; ^5^ PVE indicates phenotypic variation explained for each additive QTL; ^6^ Add indicates the additive effect of the allele in T461 genome; ^7^ Positive value in Add column indicates that T461 contributes the positive allele, and negative value in Add column indicates that WYG contributes the positive allele.

**Table 3 ijms-25-01497-t003:** The QTLs identified by the SNP-index algorithm and ED algorithm.

QTLs	Chr.	Start (bp)	End (bp)	Interval (Mb)	Peak
*bETV1*	1	20,650,001	23,510,000	2.86	0.46
*bETV2*	2	30,030,001	32,740,000	2.71	0.47
*bETV4-1*	4	21,700,001	26,760,000	5.06	0.50
*bETV4-2*	4	33,300,001	35,340,000	2.04	0.43
*bETV5*	5	15,510,001	17,590,000	2.08	0.43
*bETV7*	7	25,310,001	29,697,621	4.39	0.45
*bETV9*	9	11,840,001	14,360,000	2.52	−0.45

## Data Availability

The original contributions presented in this study are included in this article/Appendix A. The raw reads of BSA-seq and RNA-seq analysis could retrieved from the database in NCBI (https://www.ncbi.nlm.nih.gov/sra/PRJNA1018333 accessed on 18 September 2023). Further inquiries can be directed to the corresponding authors.

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
