# Peer review of "The QTL and Candidate Genes Regulating the Early Tillering Vigor Traits of Late-Season Rice in Double-Cropping Systems"

_ijms, 2024, doi:10.3390/ijms25031497_

Round 1

Reviewer 1 Report

Comments and Suggestions for Authors

ijms-2789224

Overview:

The motivation for this study is well explained, and the experimental design is adequate; however, several descriptions are insufficient.

Comments:

Spell out abbreviations.

BSA: bisulfite sequence?

T-Mix, S-Mix, TM sample, SM sample

The fluctuating descriptions below take time for readers to understand:

L19 ‘RIL population which were cultivated as double-cropping late rice in summer season’

L53 ‘Early Indica and Late Japonica’

L55-57 ‘Compared to indica, … in late-season japonica’

L92 ‘RIL population generated the crossing of a japonica rice and an indica rice’

Related to this matter, please write the line names of japonica and indica rice varieties at the end of the Introduction or at the top of the Results.

Figure 1 bottom panel

What does this picture present? Tiller angle or length? There was no explanation in the figure legend or body of the text.

Figure 3

Explanatory notes for lines and dots are not found in the figure legend.

Which plant materials did you compare to detect the DEGs? This explanation is also necessary for the text.

L183 ‘verified by qPCR’

Please cite an adequate Supplementary figure.

L232 ‘which enhances the ethylene signal’

The authors should discuss this based on RNA-seq data.

3.4 Functional unknown genes

Authors should provide BLAST search results against other grass species or Arabidopsis genomes.

Reviewer 2 Report

Comments and Suggestions for Authors

Manuscript "The QTL and candidate genes regulating the early tillering vigour traits in double-cropping late rice" by Wei Wu et al examine genetic traits that allow regulation of a valuable trait associated with tillering and double yield in rice, which currently remains poorly understood. This property is probably associated with the peculiarities of the development of organic farming.

This work made it possible to identify a number of structural features possibly related to this property. The work is interesting, although verification of these statements requires large-scale research at the cytological level and using either site-directed mutagenesis or alternative genetic engineering techniques.

In general, a manuscript can be published if changes are made. Namely, the authors should clearly outline the tasks in the introduction and, accordingly, reflect their solution in the discussion of the results and form a conclusion describing unsolved promising issues.

This critical reflection is necessary due to the fact that further development of this study is required.

In addition, it is advisable to make drawings in Fig. 2 taking into account statistics.

The same question applies to the methods of research conducting and whether this research complies with the rules of sufficient sampling.

There is no statistics and correlation section in this work, as well as a detailed description of sample preparation, which makes it difficult to evaluate this work.

I would also like to note that tables and figures in modern articles are considered as self-sufficient data, and therefore it is advisable to provide them with clearly understood full names.

I recommend supplementing the histograms with letter designations that allow you to evaluate the reliability of the differences or express them in another way.

The article can be accepted after the shortcomings have been eliminated.

Reviewer 3 Report

Comments and Suggestions for Authors

Review of " The QTL and candidate genes regulating the early tillering vigor traits in double-cropping late rice. The manuscript by Wei Wu, Ian-Tian Zhang, Li-Li You, Zi-Yi Wang, Si-Qi Du, Hai-Yan Song, Zao-Hai Wang, Ying-Jin, Huang and Jiang-Lin Liao delves into a comprehensive study aiming to identify the QTL and candidate genes regulating the early tillering vigor traits in double-cropping late rice.

The primary goal revolves around identifying the main QTLs and their candidate genes in regulating the bud to grow into rice tillers in the summer season, by carrying out QTL mapping and bulked segregant resequencing (BSA-seq) conjoint analysis using a rice recombinant inbred line (RIL) population generated the crossing of japonica rice and an indica rice. 

The manuscript effectively delineates the findings.

Some of the comments for improvement:

- Please explain all the abbreviations used in the abstract.

- RILs were used in whatever generation generations.

- BC method undesirable for getting RILs.

- Experimental Design is missing

- Please, the program used for QTL identification and linkage map.

Round 2

Reviewer 2 Report

Comments and Suggestions for Authors

The authors took into account the comments made and made changes to the text of the manuscript. Therefore, I believe that the article can be recommended for publication in its present form.